# Jensen–Inaccuracy Information Measure

**DOI:** 10.3390/e25030483

**Published:** 2023-03-10

**Authors:** Omid Kharazmi, Faezeh Shirazinia, Francesco Buono, Maria Longobardi

**Affiliations:** 1Department of Statistics, Faculty of Mathematical Sciences, Vali-e-Asr University of Rafsanjan, Rafsanjan 7718897111, Iran; 2Dipartimento di Matematica e Applicazioni “Renato Caccioppoli”, Università degli Studi di Napoli Federico II, 80138 Naples, Italy; 3Dipartimento di Biologia, Università degli Studi di Napoli Federico II, 80138 Naples, Italy

**Keywords:** Jensen–inaccuracy measure, inaccuracy measure, optimal information, Rényi entropy, average entropy, 62B10, 94A17

## Abstract

The purpose of the paper is to introduce the Jensen–inaccuracy measure and examine its properties. Furthermore, some results on the connections between the inaccuracy and Jensen–inaccuracy measures and some other well-known information measures are provided. Moreover, in three different optimization problems, the arithmetic mixture distribution provides optimal information based on the inaccuracy information measure. Finally, two real examples from image processing are studied and some numerical results in terms of the inaccuracy and Jensen–inaccuracy information measures are obtained.

## 1. Introduction

In recent decades, several researchers have studied information theory and its applications in various fields such as statistics, physics, economics, and engineering. Shannon entropy [1] is a fundamental quantity in information theory and it is defined for a continuous random variable *X* having probability density function (PDF) *f* on support X as
(1)$$\begin{array}{c} H\left(X\right)=-\int_{X}f\left(x\right)\log f\left(x\right)dx, \end{array}$$
where log denotes the natural logarithm. Throughout the paper, the support will be omitted in all the integrals. Several extensions of Shannon entropy have been considered by many researchers (see, for instance, Rényi and Tsallis entropies [2,3]) by providing one-parameter families of entropy measures.

Moreover, several divergence measures based on entropy have been defined in order to measure similarity and dissimilarity between two density functions. Among them, chi-square, Kullback–Leibler, Rényi divergences, and their extensions have been introduced; for further details, see Nielsen and Nock [4], Di Crescenzo and Longobardi [5], and Van Erven and Harremos [6].

Let *X* and *Y* be two random variables with density functions *f* and *g*, respectively. Then, the Kullback–Leibler divergence [7] is defined as
(2)$$\begin{array}{c} KL\left(f,g\right)=\int f\left(x\right)\log\frac{f(x)}{g(x)}dx, \end{array}$$
provided the integral exists. Because of some limitations of Shannon entropy, Kerridge [8] proposed a measure, known as the inaccuracy measure (or the Kerridge measure). Consider *f* and *g* as two probability density functions. Then, the inaccuracy measure between *f* and *g* is given by
$$\begin{array}{c} K(f,g)=H(f)+KL(f,g)=-\int f(x)\log g(x)dx. \end{array}$$

Several extensions of the inaccuracy measure have been developed, as well as Shannon entropy. For more details, see Kayal and Sunoj [9] and the references therein.

In the literature, the class of Jensen divergences has been studied in an extensive way as a general technique in developing information measures. Recently, other information measures such as the Jensen–Shannon, Jensen–Fisher, and Jensen–Gini measures have been studied as generalizations of well-known quantities. For further details, see Lin [10], Sánchez-Moreno et al. [11], and Mehrali et al. [12].

However, the link between the inaccuracy measure and the Jensen concept has remained unknown so far. Therefore, the main motivation in this paper is to present the Jensen–inaccuracy information (JII) measure and its properties. Let us remark that the introduction of JII measure is motivated by the fact that it can be expressed as mixture of well-known divergence measures, and it is close to the arithmetic–geometric divergence measure. Nevertheless, our new measure obtains better results when studying the similarity between elements in the field of image quality assessment. Furthermore, we establish some results associated with the connection between inaccuracy and Jensen–inaccuracy information measures and some other measures of discrimination such as Rényi entropy, average entropy, and Rényi divergence. Next, we show that the arithmetic mixture distribution provides optimal information under three different optimization problems based on the inaccuracy information measure. In the following, some well-known and useful information measures are recalled.

An extended version of the Shannon entropy measure for α>0 and α≠1, is defined by Rényi [2] as
(3)$$\begin{array}{c} R_{\alpha }\left(f\right)=\frac{\log\left(\int_{X}f^{\alpha }\left(x\right)dx\right)}{1-\alpha }. \end{array}$$

Several applications of Rényi entropy have been discussed in the literature.

Furthermore, the Rényi divergence of order α>0 and α≠1 between density functions *f* and *g* is defined by
(4)$$\begin{array}{c} D_{\alpha }\left(f,g\right)=\frac{\log\left(\int f^{\alpha }\left(x\right)g^{1-\alpha }\left(x\right)dx\right)}{\alpha -1}. \end{array}$$

The information measures in (Equation 3) and (Equation 4) become Shannon entropy and Kullback–Leibler divergence measures, respectively, when α tends to 1.

Another important diversity measure between two continuous density functions *f* and *g* is the chi-square divergence, defined as
(5)$$\begin{array}{c} \chi^{2}\left(f,g\right)=\int \frac{{\left(f(x)-g(x)\right)}^{2}}{f(x)}dx. \end{array}$$

In a similar manner, we can define $\chi^{2}\left(g,f\right)$.

The rest of this paper is organized as follows. In Section 2, we first introduce the Jensen–inaccuracy information (JII) measure. Then, we show that JII can be expressed as a mixture of the Kullback–Leibler divergence measures. We show that the Jensen–inaccuracy information measure has a close connection with the arithmetic–geometric divergence measure. Furthermore, we present an upper bound for the JII measure in terms of chi-square divergence measures. The (w,α)-Jensen–inaccuracy measure is also introduced in this section as an extended version of JII. We study the inaccuracy information measure for the escort and generalized escort distributions in Section 3. In Section 4, we consider the average entropy and define the average inaccuracy measure. Furthermore, some results are given in this regard. In Section 5, we show that the arithmetic mixture distribution involves optimal information under three different optimization problems in terms of the inaccuracy measure. Then, in Section 6, a real example is presented in order to study a problem in image processing, and some numerical results are presented in terms of the inaccuracy and Jensen–inaccuracy information measures. More precisely, our measure is useful for detection of similarity between images. Finally, in Section 7, concluding remarks are provided.

## 2. The Jensen–Inaccuracy Measure

In this section, we introduce the Jensen–inaccuracy measure and then provide a representation for this information measure in terms of Kullback–Leibler divergence measure. Furthermore, we explore the possible connection between Jensen–inaccuracy and arithmetic–geometric divergence measures. We also provide an upper bound for the Jensen–inaccuracy measure based on chi-square divergence measure. At the end of this section, we introduce (w,α)-Jensen–inaccuracy and establish a result for this extended measure.

**Definition 1.** 
*Let $f,f_{0}$, and $f_{1}$ be three density functions. Then, the Jensen–inaccuracy measure between $f_{0}$ and $f_{1}$ with respect to f is defined by*

(6)
$$\begin{array}{c} \mathrm{JK}\left(f,f_{0},f_{1}\right)=\frac{1}{2}K\left(f,f_{0}\right)+\frac{1}{2}K\left(f,f_{1}\right)-K\left(f,\frac{f_{0}+f_{1}}{2}\right). \end{array}$$



**Theorem 1.** 
*The $\mathrm{JK}(f,f_{0},f_{1})$ inaccuracy measure in (Equation 6) is non-negative.*


**Proof.** From the convexity properties of −logx,x>0, function we have
(7)$$\begin{array}{c} -\log\left(\frac{f_{0}\left(x\right)+f_{1}\left(x\right)}{2}\right)\leq -\frac{1}{2}\log f_{0}\left(x\right)-\frac{1}{2}\log f_{1}\left(x\right). \end{array}$$Now, by multiplying both sides of (Equation 7) by f(x) and then integrating with respect to *x*, we obtain
$$\begin{array}{c} -\int f\left(x\right)\log\left(\frac{f_{0}\left(x\right)+f_{1}\left(x\right)}{2}\right)dx\leq -\int f\left(x\right)\frac{1}{2}\log f_{0}\left(x\right)dx-\int f\left(x\right)\frac{1}{2}\log f_{1}\left(x\right)dx, \end{array}$$
as required. □

### 2.1. The Jensen–Inaccuracy Measure and its Connection to Kullback–Leibler Divergence

Here, we provide a representation for the Jensen–inaccuracy measure in terms of Kullback–Leibler divergence measure.

**Theorem 2.** 
*A representation for the Jensen–inaccuracy measure in (Equation 6) based on mixture of Kullback–Liebler divergence measures is given by*

(8)
$$\begin{array}{c} \mathrm{JK}\left(f,f_{0},f_{1}\right)=\frac{1}{2}KL\left(f,f_{0}\right)+\frac{1}{2}KL\left(f,f_{1}\right)-KL\left(f,\frac{f_{0}+f_{1}}{2}\right). \end{array}$$



**Proof.** According to the definition of the Jensen–inaccuracy measure and the relations
$$\begin{array}{c} K\left(f,f_{0}\right)=KL\left(f,f_{0}\right)+H\left(f\right), \end{array}$$
$$\begin{array}{c} K\left(f,f_{1}\right)=KL\left(f,f_{1}\right)+H\left(f\right), \end{array}$$
$$\begin{array}{c} K\left(f,\frac{f_{0}+f_{1}}{2}\right)=KL\left(f,\frac{f_{0}+f_{1}}{2}\right)+H\left(f\right), \end{array}$$
we have
$$\begin{array}{ccc} \mathrm{JK}(f,f_{0},f_{1}) & = & \frac{1}{2}K\left(f,f_{0}\right)+\frac{1}{2}K\left(f,f_{1}\right)-K\left(f,\frac{f_{0}+f_{1}}{2}\right)\\ & = & \frac{KL\left(f,f_{0}\right)+H\left(f\right)}{2}+\frac{KL\left(f,f_{1}\right)+H\left(f\right)}{2}-KL\left(f,\frac{f_{0}+f_{1}}{2}\right)-H\left(f\right)\\ & = & \frac{1}{2}KL\left(f,f_{0}\right)+\frac{1}{2}KL\left(f,f_{1}\right)-KL\left(f,\frac{f_{0}+f_{1}}{2}\right), \end{array}$$
as required. □

Next, we extend the definition of the Jensen–inaccuracy measure based on n+1 density functions.

**Definition 2.** 
*Let $X_{1},\ldots ,X_{n}$, and Y be random variables with density functions $f_{1},\ldots ,f_{n}$, and f, respectively, and $\alpha_{1},\ldots ,\alpha_{n}$ be non-negative real numbers such that $\sum_{i=1}^{n}\alpha_{i}=1$. Then, the Jensen–inaccuracy measure is defined as*

(9)
$$\begin{array}{c} {\mathrm{JK}}_{\alpha }\left(f,f_{1},\ldots ,f_{n}\right)=\sum_{i=1}^{n}\alpha_{i}K\left(f,f_{i}\right)-K\left(f,\sum_{i=1}^{n}\alpha_{i}f_{i}\right). \end{array}$$



**Theorem 3.** 
*The ${\mathrm{JK}}_{\alpha }\left(f,f_{1},\ldots ,f_{n}\right)$ information measure in (Equation 9) can be written in terms of Kullback–Leibler divergence as*

(10)
$$\begin{array}{c} {\mathrm{JK}}_{\alpha }\left(f,f_{1},\ldots ,f_{n}\right)=\sum_{i=1}^{n}\alpha_{i}KL\left(f,f_{i}\right)-KL\left(f,\sum_{i=1}^{n}\alpha_{i}f_{i}\right). \end{array}$$



**Proof.** From the definition of ${\mathrm{JK}}_{\alpha }\left(f,f_{1},\ldots ,f_{n}\right)$ in (Equation 9), we have
$$\begin{array}{ccc} {\mathrm{JK}}_{\alpha }\left(f,f_{1},...,f_{n}\right) & = & \sum_{i=1}^{n}\alpha_{i}K\left(f,f_{i}\right)-K\left(f,\sum_{i=1}^{n}\alpha_{i}f_{i}\right)\\ & = & \sum_{i=1}^{n}\alpha_{i}\left(KL\left(f,f_{i}\right)+H\left(f\right)\right)-KL\left(f,\sum_{i=1}^{n}\alpha_{i}f_{i}\right)-H\left(f\right)\\ & = & \sum_{i=1}^{n}\alpha_{i}KL\left(f,f_{i}\right)-KL\left(f,\sum_{i=1}^{n}\alpha_{i}f_{i}\right), \end{array}$$
as required. □

### 2.2. Connection between Jensen–Inaccuracy and Arithmetic–Geometric Divergence Measures

Now, we explore the connection between the Jensen–inaccuracy and arithmetic–geometric divergence measures. Then, we provide an upper bound for the Jensen–inaccuracy measure based on chi-square divergence measure.

Let $f_{0}$ and $f_{1}$ be two density functions. Then, the arithmetic–geometric divergence measure is defined as
(11)$$\begin{array}{c} T\left(f_{0},f_{1}\right)=\int \left(\frac{f_{0}\left(x\right)+f_{1}\left(x\right)}{2}\right)\log\left(\frac{f_{0}\left(x\right)+f_{1}\left(x\right)}{2\sqrt{f_{0}\left(x\right)f_{1}\left(x\right)}}\right)dx. \end{array}$$

For more details, see Taneja [13].

In the following definition, we provide an extension of the arithmetic–geometric divergence measure to *n*-density functions.

**Definition 3.** 
*Let $X_{1},\ldots ,X_{n}$ be random variables with density functions $f_{1},\ldots ,f_{n}$, respectively, and $\alpha_{1},\ldots ,\alpha_{n}$ be non-negative real numbers such that $\sum_{i=1}^{n}\alpha_{i}=1$. Then, the extended arithmetic–geometric divergence measure is defined as*

(12)
$$\begin{array}{c} T\left(f_{1},...,f_{n};\alpha \right)=\int f_{T}\left(x\right)\log\left(\frac{f_{T}\left(x\right)}{\prod_{i=1}^{n}f_{i}^{\alpha_{i}}\left(x\right)}\right)dx, \end{array}$$

*where $f_{T}\left(x\right)=\sum_{i=1}^{n}\alpha_{i}f_{i}\left(x\right)$ and α is a brief notation to denote $\alpha_{1},\cdots ,\alpha_{n}$.*


In the following, we explore the connection between the Jensen–inaccuracy measure in (Equation 9) and the arithmetic–geometric divergence measure in (Equation 12).

**Theorem 4.** 
*If $f=f_{T}$, then, we have*

(13)
$$\begin{array}{c} {\mathrm{JK}}_{\alpha }\left(f,f_{1},\ldots ,f_{n}\right)=T\left(f_{1},\ldots ,f_{n};\alpha \right), \end{array}$$

*where $T(f_{1},\ldots ,f_{n};\alpha )$ is the extended arithmetic–geometric divergence measure defined in (Equation 12) and α is a brief notation to denote $\alpha_{1},\cdots ,\alpha_{n}$.*


**Proof.** From the assumption $f=f_{T}$ and Theorem 3, we have
$$\begin{array}{ccc} {\mathrm{JK}}_{\alpha }\left(f_{T},f_{1},...,f_{n}\right) & = & \sum_{i=1}^{n}\alpha_{i}KL\left(f_{T},f_{i}\right)\\ & = & \sum_{i=1}^{n}\alpha_{i}\int f_{T}\left(x\right)\log\left(\frac{f_{T}\left(x\right)}{f_{i}\left(x\right)}\right)dx\\ & = & \int f_{T}\left(x\right)\log\left[\prod_{i=1}^{n}{\left(\frac{f_{T}\left(x\right)}{f_{i}\left(x\right)}\right)}^{\alpha_{i}}\right]dx\\ & = & \int f_{T}\left(x\right)\log\left(\frac{f_{T}\left(x\right)}{\prod_{i=1}^{n}f_{i}^{\alpha_{i}}\left(x\right)}\right)dx\\ & = & T(f_{1},...,f_{n};\alpha ). \end{array}$$
as required. □

**Remark 1.** 
*From the inequality logx≤x−1 for x>0, it is easy to obtain an upper bound for ${\mathrm{JK}}_{\alpha }\left(f_{T},f_{1},\ldots ,f_{n}\right)$ based on the chi-square divergence measure:*

$$\begin{array}{c} {\mathrm{JK}}_{\alpha }\left(f_{T},f_{1},...,f_{n}\right)=\sum_{i=1}^{n}\alpha_{i}KL\left(f_{T},f_{i}\right)\leq \sum_{i=1}^{n}\alpha_{i}\chi^{2}\left(f_{i},f_{T}\right). \end{array}$$



### 2.3. The (w,α)-Jensen–Inaccuracy Measure

Here, (p,w)-Jensen–inaccuracy is defined. Moreover, we establish a result for this extended measure.

**Definition 4.** 
*Let f, $f_{0}$ and $f_{1}$ be three density functions. Then, the (w,α)-Jensen–inaccuracy measure between $f_{0}$ and $f_{1}$ with respect to f is defined by*

$$\begin{array}{c} {\mathrm{JK}}_{w,\alpha }\left(f,f_{0},f_{1}\right)=wK\left(f,\left(1-p\right)f_{0}+pf_{1}\right)+\left(1-w\right)K\left(f,pf_{0}+\left(1-p\right)f_{1}\right)-K\left(f,\left(1-\bar{p}\right)f_{0}+\bar{p}f_{1}\right), \end{array}$$

*where $\bar{p}=wp+\left(1-w\right)\left(1-p\right)$.*


Note that
$$\begin{array}{c} \left(1-\bar{p}\right)f_{0}\left(x\right)+\bar{p}f_{1}\left(x\right)=w\left(\left(1-p\right)f_{0}\left(x\right)+pf_{1}\left(x\right)\right)+\left(1-w\right)\left(pf_{0}\left(x\right)+\left(1-p\right)f_{1}\left(x\right)\right). \end{array}$$

**Theorem 5.** 
*A representation for (w,α)-Jensen–inaccuracy measure based on Kullback–Leibler divergence is given by*

$$\begin{array}{ccc} {\mathrm{JK}}_{w,\alpha }\left(f,f_{0},f_{1}\right) & = & wKL\left(f,\left(1-p\right)f_{0}+pf_{1}\right)+\left(1-w\right)KL\left(f,pf_{0}+\left(1-p\right)f_{1}\right)\\ & & -KL(f,\left(1-\bar{p}\right)f_{0}+\bar{p}f_{1}), \end{array}$$



**Proof.** It can be proven in the same manner as Theorem 2. □

## 3. Inaccuracy Information Measure of the Escort and Generalized Escort Distributions

The escort distribution is a baseline definition in non-extensive statistical mechanics and coding theory; it is closely related to Tsallis and Rényi entropies. For more details, see Bercher [14]. We show that the inaccuracy measure between an arbitrary density and its corresponding escort density can be expressed as a mixture of Shannon and Rényi entropies. Furthermore, another finding associated with the inaccuracy measure between a generalized escort distribution and each of its components reveals some interesting connections in terms of Kullback–Leibler and Rényi divergences.

Let *f* be a density function. Then, the escort density with order α>0 associated with *f* is defined as
(14)$$\begin{array}{c} f_{\alpha }\left(x\right)=\frac{f^{\alpha }\left(x\right)}{\int f^{\alpha }\left(x\right)dx}. \end{array}$$
**Theorem 6.** *Let f be a density function and $f_{\alpha }$ be an escort density corresponding to f. Then, for α>0, we obtain:*(i)*$K\left(f,f_{\alpha }\right)=\alpha H\left(f\right)+\left(1-\alpha \right)R_{\alpha }\left(f\right)$;*(ii)*$K\left(f_{\alpha },f\right)=\frac{1}{\alpha }H\left(f_{\alpha }\right)+\frac{\alpha -1}{\alpha }R_{\alpha }\left(f\right)$,**where $R_{\alpha }\left(f\right)$ is Rényi entropy in (Equation 3).*
**Proof.** From the definition of inaccuracy measure between *f* and $f_{\alpha }$, we have
$$\begin{array}{ccc} K(f,f_{\alpha }) & = & -\int f\left(x\right)\log f_{\alpha }\left(x\right)dx=-\int f\left(x\right)\log\left(\frac{f^{\alpha }\left(x\right)}{\int f^{\alpha }\left(x\right)dx}\right)dx\\ & = & -\alpha \int f\left(x\right)\log f\left(x\right)dx+\int f\left(x\right)\log\left(\int f^{\alpha }\left(x\right)dx\right)dx\\ & = & \alpha H\left(f\right)+\log\left(\int f^{\alpha }\left(x\right)dx\right)\\ & = & \alpha H\left(f\right)+\left(1-\alpha \right)R_{\alpha }\left(f\right), \end{array}$$
which proves (i). Next
$$\begin{array}{ccc} K(f_{\alpha },f) & = & -\int f_{\alpha }\left(x\right)\log f\left(x\right)dx=-\int \left(\frac{f^{\alpha }\left(x\right)}{\int f^{\alpha }\left(x\right)dx}\right)\log f\left(x\right)dx\\ & = & -\frac{1}{\alpha }\int \left(\frac{f^{\alpha }\left(x\right)}{\int f^{\alpha }\left(x\right)dx}\right)\log f^{\alpha }\left(x\right)dx\\ & = & -\frac{1}{\alpha }\int \left(\frac{f^{\alpha }\left(x\right)}{\int f^{\alpha }\left(x\right)dx}\right)\log\left(\frac{f^{\alpha }\left(x\right)}{\int f^{\alpha }\left(x\right)dx}\right)dx\\ & & -\frac{1}{\alpha }\int \left(\frac{f^{\alpha }\left(x\right)}{\int f^{\alpha }\left(x\right)dx}\right)\log\left(\int f^{\alpha }\left(x\right)dx\right)dx\\ & = & \frac{1}{\alpha }H\left(f_{\alpha }\right)+\frac{\alpha -1}{\alpha }R_{\alpha }\left(f\right), \end{array}$$
which proves (ii). □

Let *f* and *g* be two probability density functions. Then, the generalized escort density for α∈(0,1) is defined as
(15)$$\begin{array}{c} h_{\alpha }\left(x\right)=\frac{f^{\alpha }\left(x\right)g^{1-\alpha }\left(x\right)}{\int f^{\alpha }\left(x\right)g^{1-\alpha }\left(x\right)dx}. \end{array}$$

**Theorem 7.** 
*The inaccuracy information measure between f and the generalized escort density $h_{\alpha }$ is given by*

(16)
$$\begin{array}{c} K\left(f,h_{\alpha }\right)=\left(\alpha -1\right)D_{\alpha }\left(f,g\right)-\alpha KL\left(f,g\right)-K\left(f,g\right), \end{array}$$

*where $D_{\alpha }\left(f,g\right)$ is the relative Rényi entropy defined by*

(17)
$$\begin{array}{c} D_{\alpha }\left(f,g\right)=\frac{\log\left(\int f^{\alpha }\left(x\right)g^{1-\alpha }\left(x\right)dx\right)}{\alpha -1}. \end{array}$$



**Proof.** From the definition of $K(f,h_{\alpha })$, we derive
$$\begin{array}{ccc} K(f,h_{\alpha }) & = & -\int f\left(x\right)\log h_{\alpha }\left(x\right)dx=-\int f\left(x\right)\log\left(\frac{f^{\alpha }\left(x\right)g^{1-\alpha }\left(x\right)}{\int f^{\alpha }\left(x\right)g^{1-\alpha }\left(x\right)dx}\right)dx\\ & = & -\int f\left(x\right)\log{\left(\frac{f(x)}{g(x)}\right)}^{\alpha }dx+\int f\left(x\right)\log\left(\int f^{\alpha }\left(x\right)g^{1-\alpha }\left(x\right)dx\right)dx-K\left(f,g\right)\\ & = & -\alpha \int f\left(x\right)\log\frac{f(x)}{g(x)}dx+\log\left(\int f^{\alpha }\left(x\right)g^{1-\alpha }\left(x\right)dx\right)-K\left(f,g\right)\\ & = & \left(\alpha -1\right)D_{\alpha }\left(f,g\right)-\alpha KL\left(f,g\right)-K\left(f,g\right), \end{array}$$
as required. □

**Theorem 8.** 
*Let $f_{0}$ and $f_{1}$ be two density functions and consider the arithmetic and geometric mixture densities, respectively, as $f_{a}\left(x\right)=pf_{0}\left(x\right)+\left(1-p\right)f_{1}\left(x\right)$ and $f_{g}\left(x\right)=\frac{f_{0}^{p}\left(x\right)f_{1}^{1-p}\left(x\right)}{\int f_{0}^{p}\left(x\right)f_{1}^{1-p}\left(x\right)dx}.$ Then, a lower bound for $K(f_{a},f_{g})$ is given by*

(18)
$$\begin{array}{c} K\left(f_{a},f_{g}\right)\geq H\left(f_{a}\right)+\left(1-p\right)D_{p}\left(f_{0},f_{1}\right). \end{array}$$



**Proof.** From the definition $K(f_{a},f_{g})$, by using the arithmetic mean–geometric mean inequality, we have
$$\begin{array}{ccc} K(f_{a},f_{g}) & = & -\int f_{a}\left(x\right)\log f_{g}\left(x\right)dx=-\int \left(pf_{0}\left(x\right)+\left(1-p\right)f_{1}\left(x\right)\right)\log\left(\frac{f_{0}^{p}\left(x\right)f_{1}^{1-p}\left(x\right)}{\int f_{0}^{p}\left(x\right)f_{1}^{1-p}\left(x\right)dx}\right)dx\\ & = & -\int \left(pf_{0}\left(x\right)+\left(1-p\right)f_{1}\left(x\right)\right)\log\left(f_{0}^{p}\left(x\right)f_{1}^{1-p}\left(x\right)\right)dx-\log\left(\int f_{0}^{p}\left(x\right)f_{1}^{1-p}\left(x\right)dx\right)\\ & \geq & -\int \left(pf_{0}\left(x\right)+\left(1-p\right)f_{1}\left(x\right)\right)\log\left(pf_{0}\left(x\right)+\left(1-p\right)f_{1}\left(x\right)\right)dx-\log\left(\int f_{0}^{p}\left(x\right)f_{1}^{1-p}\left(x\right)dx\right)\\ & = & H\left(pf_{0}+\left(1-p\right)f_{1}\right)+\left(1-p\right)D_{p}\left(f_{0},f_{1}\right)\\ & = & H\left(f_{a}\right)+\left(1-p\right)D_{p}\left(f_{0},f_{1}\right), \end{array}$$
as required. □

## 4. Inaccuracy Measure Based on Average Entropy

Let *X* be a random variable with PDF *f*. Then, the average entropy associated with *f* is defined as
(19)$$\begin{array}{c} AE\left(f\right)=-\int f\left(x\right)\log\left(\frac{f(x)}{\int f^{2}\left(x\right)dx}\right)dx \end{array}$$

For pertinent details, see Kittaneh et al. [15].

**Theorem 9.** 
*Let f be a density function. Then, an upper bound for the Shannon entropy based on inaccuracy measure is given by*

(20)
$$\begin{array}{c} H\left(f\right)\leq K\left(f,f_{2}\right), \end{array}$$

*where $f_{2}$ is the corresponding escort distribution of the density f with order 2.*


**Proof.** From the definition of average entropy, we have
$$\begin{array}{ccc} AE(f) & = & -\int f\left(x\right)\log\left(\frac{f(x)}{\int f^{2}\left(x\right)dx}\right)dx\\ & = & -\int f\left(x\right)\log\left(\frac{f^{2}\left(x\right)}{\int f^{2}\left(x\right)dx}\right)dx+\int f\left(x\right)\log f\left(x\right)dx\\ & = & K\left(f,f_{2}\right)-H\left(f\right). \end{array}$$Now, because AE(f) is non-negative (see Theorem 1 of Kittaneh et al. [15]), we have
(21)$$\begin{array}{c} K\left(f,f_{2}\right)\geq H\left(f\right), \end{array}$$
as required. □

**Definition 5.** 
*Let X be a random variable with density f. Then, the average inaccuracy measure is defined as*

(22)
$$\begin{array}{c} \mathrm{AK}\left(f,g\right)=-\int f\left(x\right)\log\left(\frac{g(x)}{\int g^{2}\left(x\right)dx}\right)dx. \end{array}$$



**Theorem 10.** 
*Let f and g be two PDFs. Then, the average inaccuracy measure between f and g, AK(f,g), can be expressed as*

(23)
$$\begin{array}{c} \mathrm{AK}\left(f,g\right)=K\left(f,g\right)-R_{2}\left(g\right). \end{array}$$



**Proof.** From the definition of the average inaccuracy measure, we have
$$\begin{array}{ccc} \mathrm{AK}(f,g) & = & -\int f\left(x\right)\log\left(\frac{g(x)}{\int g^{2}\left(x\right)dx}\right)dx\\ & = & -\int f\left(x\right)\log g\left(x\right)dx+\log\left(\int g^{2}\left(x\right)dx\right)\\ & = & K\left(f,g\right)-R_{2}\left(g\right), \end{array}$$
as required. □

## 5. Optimal Information Model under Inaccuracy Information Measure

In this section, we prove that the arithmetic mixture distribution provides optimal information under three different optimization problems associated with inaccuracy information measures. For more details on optimal information properties of some statistical distributions, one may refer to Kharazmi et al. [16] and the references therein.

**Theorem 11.** 
*Let f, $f_{0}$, and $f_{1}$ be three density functions. Then, the solution to the optimization problem*

(24)
$$\begin{array}{c} \min_{f}K\left(f_{0},f\right)\;\;subject\;to\;K\left(f_{1},f\right)=\eta ,\;\;\;\;\int f\left(x\right)dx=1, \end{array}$$


*is the arithmetic mixture density with mixing parameter $p=\frac{1}{1+\lambda_{0}}$, $\lambda_{0}>0$ is the Lagrangian multiplier, and η is a constraint associated with the optimization problem.*


**Proof.** We use the Lagrangian multiplier technique in order to solve the optimization problem in (Equation 24). Thus, we have
$$\begin{array}{c} L\left(f,\lambda_{0},\lambda_{1}\right)=-\int f_{0}\left(x\right)\log f\left(x\right)dx-\lambda_{0}\int f_{1}\left(x\right)\log f\left(x\right)dx+\lambda_{1}\int f\left(x\right)dx. \end{array}$$Now, differentiating with respect to *f*, we obtain
(25)$$\begin{array}{c} \frac{\partial }{\partial f}L\left(f,\lambda_{0},\lambda_{1}\right)=-\frac{f_{0}\left(x\right)}{f(x)}-\lambda_{0}\frac{f_{1}\left(x\right)}{f(x)}+\lambda_{1}. \end{array}$$By setting (Equation 25) to zero, we derive the optimal density function as
$$\begin{array}{c} f\left(x\right)=pf_{0}\left(x\right)+\left(1-p\right)f_{1}\left(x\right), \end{array}$$
where $p=\frac{1}{1+\lambda_{0}}$, as required. In fact, we obtain the solution based on f(x) as
$$\begin{array}{c} f\left(x\right)=\frac{f_{0}\left(x\right)+\lambda_{0}f_{1}\left(x\right)}{\lambda_{1}}. \end{array}$$From the normalization condition, we have
$$\begin{array}{c} \int f\left(x\right)dx=\int \left(\frac{f_{0}\left(x\right)+\lambda_{0}f_{1}\left(x\right)}{\lambda_{1}}\right)dx=1, \end{array}$$
and then $\lambda_{1}=1+\lambda_{0}$. □

**Theorem 12.** 
*Let f, $f_{0}$, and $f_{1}$ be three density functions. Then, the solution to the optimization problem,*

(26)
$$\begin{array}{c} \min_{f}\left\{wK\left(f_{0},f\right)+\left(1-w\right)K\left(f_{1},f\right)\right\}\;\;subject\;to\;\int f\left(x\right)dx=1,\;0\leq w\leq 1, \end{array}$$


*is the arithmetic mixture density with mixing parameter p=w.*


**Proof.** Making use of the Lagrangian multiplier technique as in Theorem 11, the result follows. □

**Theorem 13.** 
*Let f, $f_{0}$, and $f_{1}$ be three density functions and $T_{\alpha }\left(X\right)=\frac{f_{0}\left(X\right)}{f^{2}\left(X\right)}$. Then, the solution to the optimization problem,*

(27)
$$\begin{array}{c} \min_{f}K\left(f_{0},f\right)\;\;subject\;to\;E_{f}\left(T_{\alpha }\left(X\right)\right)=\eta ,\;\;\;\;\int f\left(x\right)dx=1, \end{array}$$


*is the arithmetic mixture density with mixing parameter $p=\frac{1}{1+\lambda_{0}}$, $\lambda_{0}>0$ is the Lagrangian multiplier, $E_{f}\left(\cdot \right)$ is the expectation with respect to f and η is a constraint associated with the optimization problem.*


**Proof.** The result follows analogously with the proof of Theorem 11. □

## 6. Application

In this section, we first consider the definition of histogram for a given image in the context of image quality assessment. Then, we illustrate two applications by using the inaccuracy and Jensen–inaccuracy measures.

### 6.1. Image and Histogram

A digital image is defined as a discrete set of small surface elements (pixel). One such digital image is a grayscale image in which each pixel only contains one value (its intensity). These values are in the set {0,…,L−1}, where *L* represents the number of intensity values. Suppose that $n_{k}$ is the number of times in which the *k*th intensity appears in the image. Furthermore, the corresponding histogram of a digital image refers to a histogram of the pixel intensity values in the set {0,…,L−1} (one may refer to Gonzalez [17]).

### 6.2. Non-Parametric Jensen–Inaccuracy Estimation

We now show an application of the inaccuracy and Jensen–inaccuracy measures defined in (Equation 6) to image processing. Let $X_{1},\cdots ,X_{n}$ be a random sample with probability density function *f*. Then, the kernel estimate of density *f* based on kernel function *K* with bandwidth $h_{X}>0$ at a fixed point *x* is given by
(28)$$\begin{array}{c} \hat{f}\left(x\right)=\frac{1}{nh_{X}}\sum_{i=1}^{n}K\left(\frac{x-X_{i}}{h_{X}}\right). \end{array}$$

Similarly, the non-parametric estimate of density *g* with bandwidth $h_{Y}>0$, based on random sample $Y_{1},\cdots ,Y_{n}$, is expressed as
(29)$$\begin{array}{c} \hat{g}\left(x\right)=\frac{1}{nh_{Y}}\sum_{i=1}^{n}K\left(\frac{x-Y_{i}}{h_{Y}}\right). \end{array}$$

For more details, see Duong et al. [18]. Upon making use of (Equation 28) and (Equation 29), the integrated non-parametric estimate of the inaccuracy and Jensen–inaccuracy measures are given, respectively, by
$$\begin{array}{ccc} \hat{K}\left(f,g\right) & = & -\int \hat{f}\left(x\right)\log\hat{g}\left(x\right)dx\\ & = & -\int \frac{1}{nh_{X}}\sum_{i=1}^{n}K\left(\frac{x-X_{i}}{h_{X}}\right)\log\left(\frac{1}{nh_{Y}}\sum_{i=1}^{n}K\left(\frac{x-Y_{i}}{h_{Y}}\right)\right)dx \end{array}$$
and
$$\begin{array}{ccc} \hat{\mathrm{JK}}\left(f,f_{0},f_{1}\right) & = & \frac{1}{2}K\left(\hat{f},\hat{f_{0}}\right)+\frac{1}{2}K\left(\hat{f},\hat{f_{1}}\right)-K\left(\hat{f},\frac{\hat{f_{0}}+\hat{f_{1}}}{2}\right)\\ & = & -\int \frac{1}{nh}\sum_{i=1}^{n}K\left(\frac{x-X_{i}}{h}\right)\log\left(\frac{1}{nh_{0}}\sum_{i=1}^{n}K\left(\frac{x-Y_{i}}{h_{0}}\right)\right)dx\\ & & -\int \frac{1}{nh}\sum_{i=1}^{n}K\left(\frac{x-X_{i}}{h}\right)\log\left(\frac{1}{nh_{1}}\sum_{i=1}^{n}K\left(\frac{x-Y_{i}}{h_{1}}\right)\right)dx\\ & & +\int \frac{1}{nh}\sum_{i=1}^{n}K\left(\frac{x-X_{i}}{h}\right)\log\left(\frac{\frac{1}{nh_{0}}\sum_{i=1}^{n}K\left(\frac{x-Y_{i}}{h_{0}}\right)+\frac{1}{nh_{1}}\sum_{i=1}^{n}K\left(\frac{x-Y_{i}}{h_{1}}\right)}{2}\right)dx, \end{array}$$
where $h_{0}$, $h_{1}$ and *h* are the corresponding bandwidths for the kernel estimations of the densities $f_{0}$, $f_{1}$ and *f*, respectively. Here we use Gaussian kernel $K\left(u\right)=\frac{1}{\sqrt{2\pi }}e^{-\frac{u^{2}}{2}}$.

Next, we present two examples of image processing (two reference images including grayscale cameraman and lake images) and compute the inaccuracy and Jensen–inaccuracy information measures between the original picture and each of its adjusted versions for both cases.
**Cameraman image**

Figure 1 shows the original cameraman picture denoted by *X* and three adjusted versions of this original picture considered as Y(=X+0.3) (increasing brightness), Z(=2×X) (increasing contrast), and W(=0.5×X+0.5) (increasing brightness and decreasing contrast). The cameraman image includes 512×512 cells and the gray level of each cell has a value between 0 (black) and 1 (white).
**Lake image**

Figure 2 shows the original lake image that includes 512×512 cells and the level of the color gray of each cell assumes a value in the interval [0,1] (0 for black and 1 for white). This image labeled as *X* and three adjusted versions of it labeled as Y(=X+0.3) (increasing brightness), Z(=2×X) (increasing contrast) and $W(=\sqrt{X})$ (gamma corrected).

Now, we first compute the inaccuracy between the original image and each of its adjusted versions. Then, we obtain the amount of the dissimilarity between each pair of the interference images with respect to original image based on the Jensen–inaccuracy information measure. For both images, we consider three interferences of the original images, as described above. For more details, see the *EBImage* package in *R* software [19].

The extracted histograms are plotted in Figure 3 and Figure 4, with the corresponding empirical densities for pictures *X*, *Y*, *Z*, and *W* for the cameraman and lake images, respectively.

We can see from Figure 1 and Figure 3 that the similarity has the highest degree related to *W* and then to *Y*, whereas *Z* has a divergence of the highest degree with respect to *X* that is the original picture. Moreover, from Figure 2 and Figure 4, the same observation is also found for the lake image and its three adjusted versions. We have presented the inaccuracy and Jensen–inaccuracy information measures (for both cameraman and lake images) for all pictures in Table 1. Therefore, the inaccuracy and Jensen–inaccuracy information measures can be considered as efficient criteria for comparing the similarity between an original picture and its adjusted versions.

According to Fan et al. [20], we have tried to observantly follow axioms 1 and 2 of axiomatic design theory, which has been proposed in recent decades; axiom 1 is about verification of the validity of designs, and according to axiom 2, one must choose the best design among several options.

## 7. Conclusions

In this paper, by considering the inaccuracy measure, we have proposed Jensen–inaccuracy and (w,α)-Jensen–inaccuracy information measures. We have specifically shown that Jensen–inaccuracy is connected to the arithmetic–geometric divergence measure. Then, we have studied the inaccuracy measure between the escort distribution and its underling density. Furthermore, we have examined the inaccuracy measure between the generalized escort distribution and its components. It has been shown that these inaccuracy measures are closely connected with Rényi entropy, average entropy, and Rényi divergence. Interestingly, we have shown that the arithmetic mixture distribution provides optimal information under three different optimization problems associated with the inaccuracy measure. Finally, we have described two applications of the inaccuracy and Jensen–inaccuracy measures to image processing. We have considered three adjusted versions of the original cameraman and lake images and then have examined the dissimilarity between the original image and each of its adjusted versions for both cases.

## Figures and Tables

**Figure 1 entropy-25-00483-f001:**
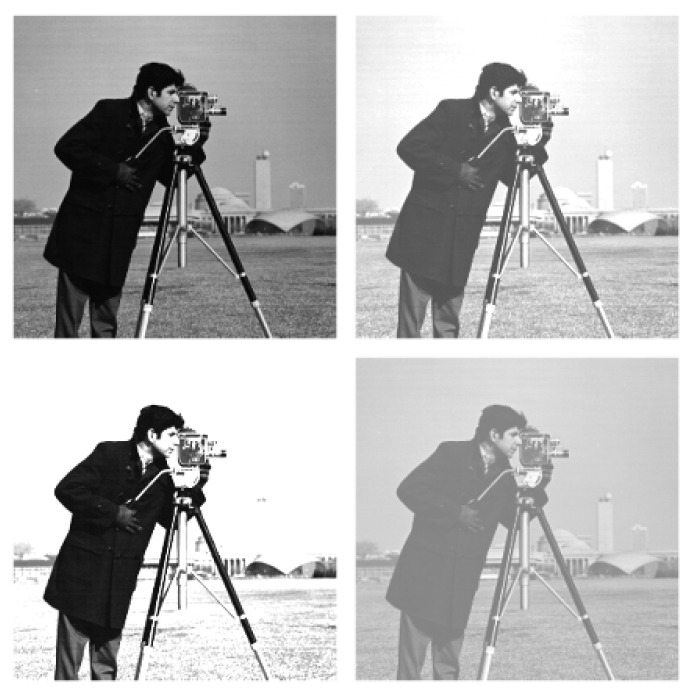
The cameraman image with its three adjusted versions. First row (left panel) original *X*; first row (right panel) *Y* (increasing brightness); second row (left panel) *Z* (increasing contrast); second row (right panel) *W* (increasing brightness and decreasing contrast).

**Figure 2 entropy-25-00483-f002:**
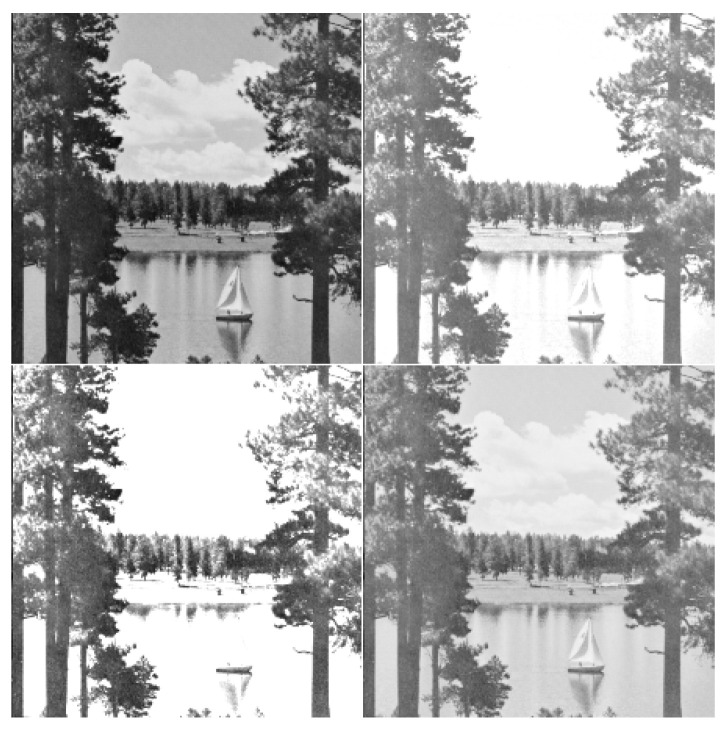
The lake image with its three adjusted versions. First row (left panel) original *X*; first row (right panel) *Y* (increasing brightness); second row (left panel) *Z* (increasing contrast); second row (right panel) *W* (gamma corrected).

**Figure 3 entropy-25-00483-f003:**
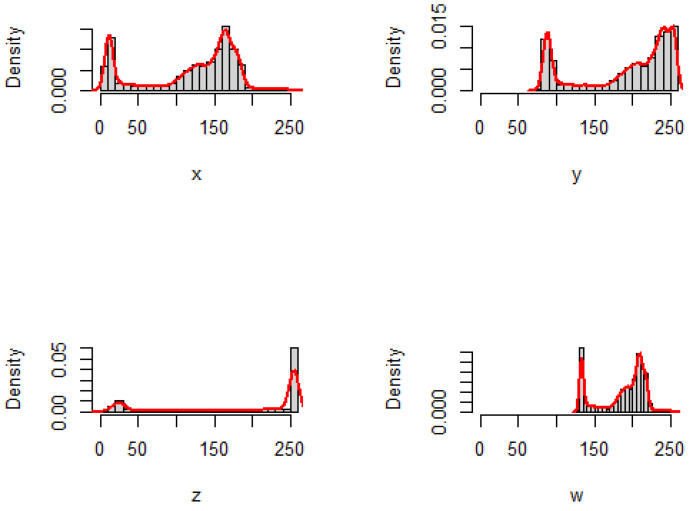
The histograms and the corresponding empirical densities for cameraman image and its three adjusted versions.

**Figure 4 entropy-25-00483-f004:**
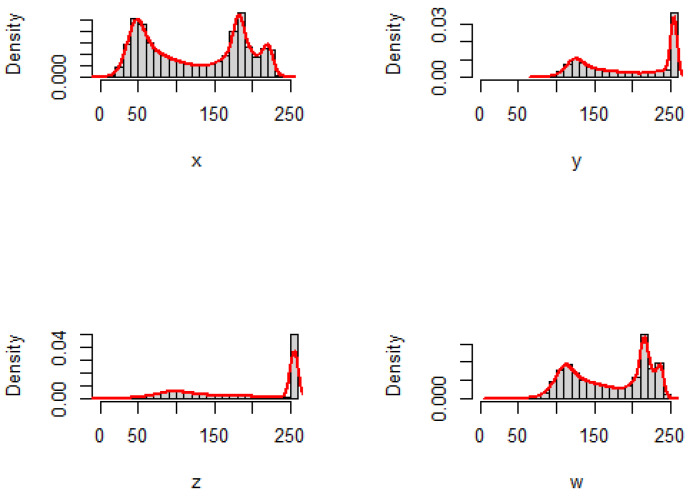
The histograms and the corresponding empirical densities for lake image and its three adjusted versions.

**Table 1 entropy-25-00483-t001:** Inaccuracy measures for the cameraman and lake images.

Cameraman Image		Lake Image	
Inaccuracy	Jensen–Inaccuracy	Inaccuracy	Jensen–Inaccuracy
X↔Y 8.6142	$\left(Y,Z\|X\right)$ 0.9674	X↔Y 10.7143	$\left(Y,Z\|X\right)$ 1.9235
X↔Z 7.3654	$\left(Y,W\|X\right)$ 0.5707	X↔Z 6.5744	$\left(Y,W\|X\right)$ 1.2558
X↔W 9.2097	$\left(Z,W\|X\right)$ 1.4315	X↔W 7.4086	$\left(Z,W\|X\right)$ 0.5131

## Data Availability

Not applicable.

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
