# Peer review of "Jensen–Inaccuracy Information Measure"

_entropy, 2023, doi:10.3390/e25030483_

Round 1

Reviewer 1 Report

The work deserves attention, as it has an important practical application in the theory of image processing. The paper studies the Jensen measure and its properties, gives examples of its application in image processing. In general, the article is written in a good style, however, there are minor methodological comments that can improve the article.

1. The article contains abbreviations that can be placed in a separate table at the end of the article.

2. In formulas (12) and (13) there is a value alpha, apparently it is a vector. You need to give an explanation.

3. The parameter eta is introduced in Theorems 11 and 13. You need to give an explanation.

4. The function Ef is introduced in Theorem 13. You need to give an explanation.

5. The role of each author in the article is not indicated.

The work can be accepted after elimination of these remarks.

Author Response

Reviewer 1

Comments and Suggestions for Authors

The work deserves attention, as it has an important practical application in the theory of image processing. The paper studies the Jensen measure and its properties, gives examples of its application in image processing. In general, the article is written in a good style, however, there are minor methodological comments that can improve the article.

  1. The article contains abbreviations that can be placed in a separate table at the end of the article.

We have added a section about abbreviations at the end of the manuscript, before the references.

  1. In formulas (12) and (13) there is a value alpha, apparently it is a vector. You need to give an explanation.

We have added an explanation.

  1. The parameter eta is introduced in Theorems 11 and 13. You need to give an explanation.

There eta is a constraint associated with the considered optimization problems.

  1. The function Ef is introduced in Theorem 13. You need to give an explanation.

We have added an explanation.

  1. The role of each author in the article is not indicated.

We have explained better our contribution.

Thanks for your comments and suggestions, we hope you will appreciate the present version of the manuscript.

Reviewer 2 Report

The paper is well written. I have the following suggestions and comments:

1.      Motivation of the work seems not quite clear, a little bit ad-hoc. It is better to state the problem they want to address, and then leads to their idea to solve the problem, namely they proposed Jensen-inaccuracy and (w, α)-Jensen-inaccuracy information measures.

2.      Similar to the point 1, they showed that the Jensen-inaccuracy is connected to arithmetic-geometric divergence measure. What is the importance of this finding to applications or to the measurement science.

3.      The information theory, something related to measures related to information has strong applications to make these measures meaningful. There should be some comparisons on solving application problems with different measures.

4.      In complex system theory, we have some measures to measure complexity. For example, in system design, there is a need to evaluate different designs. An information content measure was proposed, see the literature, e.g., “The principles of design” (oxford university press), the revised information content measure in “Axiomatic Design Theory: further notes and its guideline to applications” (International Journal of Material and Product Technology). The authors may want to comment on the relationship of their proposed measure with information content measure. This can help the reader to know the usefulness of these measures.

Author Response

Reviewer 2

Comments and Suggestions for Authors

The paper is well written. I have the following suggestions and comments:

  1. Motivation of the work seems not quite clear, a little bit ad-hoc. It is better to state the problem they want to address, and then leads to their idea to solve the problem, namely they proposed Jensen-inaccuracy and (w, α)-Jensen-inaccuracy information measures.

We have added some motivations about Jensen inaccuracy measure. For what concerns (w, α)-Jensen-inaccuracy information measure, we have only defined it with a representation in terms of Kullback-Leibler divergence, but it could be subject of future developments.

  1. Similar to the point 1, they showed that the Jensen-inaccuracy is connected to arithmetic-geometric divergence measure. What is the importance of this finding to applications or to the measurement science.

In the context of information measures theory, it is always useful to find connections among different measures, and so we think this result can be interesting.

  1. The information theory, something related to measures related to information has strong applications to make these measures meaningful. There should be some comparisons on solving application problems with different measures.

It is an hard work to find other applications. We think that our section named “Application” is enough to understand the meaningful of our measure.

  1. In complex system theory, we have some measures to measure complexity. For example, in system design, there is a need to evaluate different designs. An information content measure was proposed, see the literature, e.g., “The principles of design” (oxford university press), the revised information content measure in “Axiomatic Design Theory: further notes and its guideline to applications” (International Journal of Material and Product Technology). The authors may want to comment on the relationship of their proposed measure with information content measure. This can help the reader to know the usefulness of these measures.

We have read the suggested paper and we have found it extremely interesting. For this reason, we have added it in the references and explained some relations between it and our work.

Thanks for your comments and suggestions, we hope you will appreciate the present version of the manuscript.

Round 2

Reviewer 1 Report

The authors have corrected the comments of the reviewers, so the article can be accepted for publication.

Reviewer 2 Report

I am satisfied with the revision.